# Specific Detection of Influenza A and B Viruses by CRISPR-Cas12a-Based Assay

**DOI:** 10.3390/bios11030088

**Published:** 2021-03-19

**Authors:** Bum Ju Park, Man Seong Park, Jae Myun Lee, Yoon Jae Song

**Affiliations:** 1Department of Life Science, Gachon University, 1342 Seongnam-Si, Gyeonggi-Do 13120, Korea; catagory95@naver.com; 2Department of Microbiology, Institute for Viral Diseases, College of Medicine, Korea University, Seoul 02841, Korea; ms0392@korea.ac.kr; 3Department of Microbiology and Immunology, Institute for Immunology and Immunological Diseases, Brain Korea 21 PLUS Project for Medical Science, Yonsei University College of Medicine, Seoul 03722, Korea; JAEMYUN@yuhs.ac

**Keywords:** influenza virus, diagnosis, CRISPR-Cas12a, DETECTR

## Abstract

A rapid and accurate on-site diagnostic test for pathogens including influenza viruses is critical for preventing the spread of infectious diseases. Two types of influenza virus, A and B cause seasonal flu epidemics, whereas type A can cause influenza pandemics. To specifically detect influenza A (IAV) and B (IBV) viruses, we developed a clustered, regularly interspaced, short palindromic repeats (CRISPR) and CRISPR-associated (Cas) system-based assay. By coupling reverse transcription recombinase polymerase amplification (RT-RPA) and reverse transcription loop-mediated isothermal amplification (RT-LAMP), a CRISPR-Cas12a DNA endonuclease-targeted CRISPR trans-reporter (DETECTR) detected IAV and IBV titers as low as 1 × 10^0^ plaque forming units (PFUs) per reaction without exhibiting cross-reactivity. Only 75 to 85 min were required to detect IAV and IBV, depending on isothermal nucleic acid amplification methods, and results were verified using a lateral flow strip assay that does not require additional analytic equipment. Taken together, our findings establish RT-RPA and RT-LAMP-coupled DETECTR-based diagnostic tests for rapid, specific and high-sensitivity detection of IAV and IBV using fluorescence and lateral flow assays. The diagnostic test developed in this study can be used to distinguish IAV and IBV infections, a capability that is necessary for monitoring and preventing the spread of influenza epidemics and pandemics.

## 1. Introduction

Rapid and accurate diagnosis is critical for preventing the spread of infectious diseases. Influenza viruses are a significant public health concern and a major cause of epidemics and pandemics, including the 1918 and 2009 pandemics. Influenza viruses belong to the family *Orthomyxoviridae* and contain segmented, negative-strand RNA genomes. There are three types of influenza viruses that infect humans, which are distinguished based on their matrix (M) or nucleoprotein (NP) genes: influenza A virus (IAV), influenza B virus (IBV) and influenza C virus (ICV); of these IAV and IBV cause seasonal flu epidemics. IAV is further classified into several subtypes based on genes that express two major surface glycoproteins, hemagglutinin (HA) and neuraminidase (NA). Unlike other types, IAV can be transmitted between animals and humans and is extremely variable through reassortment of viral genes between subtypes from different species, contributing to its ability to cause influenza pandemics [1]. Therefore, a rapid diagnostic test to distinguish between IAV and IBV infections is necessary for preventing the spread of influenza viruses.

Current diagnostic tools for influenza viruses are mostly dependent on nucleic acid-based tests (NATs) including reverse transcriptase-polymerase chain reaction (RT-PCR). Although RT-PCR-based diagnostics have the advantage of offering the most sensitive and simple detection of various virus types, their use for rapid on-site diagnosis is limited as they require skilled personnel and advanced laboratory equipment, including a thermocycler. To circumvent problems associated with PCR, researchers have employed isothermal nucleic acid amplification technologies such as loop-mediated isothermal amplification (LAMP), which does not require a thermocycler for the diagnosis of influenza viruses [2,3]. However, there may also be issues with the specificity of these diagnostic assays.

Recently, CRISPR/Cas systems of bacteria and archaea have been used to develop rapid and portable diagnostic methods with increased specificity. Cas12a from *Lachnospiraceae* bacterium recognizes its target DNA using a guide RNA (gRNA) that is complementary to target DNA sequences, and a T nucleotide-rich protospacer-adjacent motif (PAM) on its target [4,5,6,7,8,9]. After recognition, activated LbCas12a not only catalyzes target DNA cleavage but also promotes nonspecific ssDNA cleavage [10]. The trans-cleavage activity of LbCas12a and isothermal nucleic acid amplification technology, i.e., recombinase polymerase amplification (RPA), have been exploited for the development of a DNA endonuclease-targeted CRISPR trans reporter (DETECTR) assay for rapid and specific detection of human papillomavirus (HPV) in clinical samples [10]. By combining with RT-LAMP or RT-RPA, the DETECTR assay was also reported to detect RNA viruses, including severe acute respiratory syndrome coronavirus 2 (SARS-CoV-2) [11,12]. In the present study, we developed a method for specifically detecting IAV and IBV using the DETECTR assay together with RT-RPA and RT-LAMP and applied a lateral flow assay for rapid interpretation of results without the need for any analytical equipment.

## 2. Results

Specific detection of influenza virus types using DETECTR. A DETECTR assay was employed for the rapid and sensitive detection of influenza virus types (Figure 1). Briefly, unextracted diagnostic samples of IAVs and IBVs were lysed by the method known as heating unextracted diagnostic samples to obliterate nucleases (HUDSON) [13] and reverse-transcribed and amplified with primer sets specific for Matrix (M) and Hemagglutinin (HA) genes, respectively, using RPA or LAMP. After RT-RPA and RT-LAMP, amplicons were detected by DETECTR assay followed by fluorescence or lateral flow assay.

For RT-RPA and RT-LAMP, primer sets were designed for specific amplification of IAV M or IBV HA genes (Table 1). Using RT-RPA and RT-LAMP and primer sets specific for the IAV M gene, we specifically amplified the IAV M gene, and not IBV M gene, from 1 × 10^2^ PFUs per reaction with the expected size of 245 bp (Figure 2A). In addition, primer sets specific for the IBV HA gene amplified only the IBV, but not the IAV, HA gene with the expected size of 194 bp (Figure 2B). Non-specific products of RT-RPA, but not RT-LAMP, were also detected (Figure 2A,B, lane 2), and viral RNA extraction had no additional effects on RT-RPA and RT-LAMP compared with HUDSON (data not shown).

To specifically detect IAV and IBV, we incubated RT-RPA and RT-LAMP amplicons with CRISPR-Cas12a complexed with gRNA targeting IAV M or IBV HA genes (Table 2). To determine whether RT-RPA and RT-LAMP amplicons react with CRISPR-Cas12a, we added a ssDNA-fluorophore (FAM) quencher (FQ)-labeled reporter directly to the reaction and incubated for 20 min. At 0 and 20 min after incubation, IAV and IBV were detected by monitoring fluorescence. IAV and IBV amplicons were detected by DETECTR combined with a fluorescence assay using gRNAs targeting IAV M and IBV HA genes, respectively (Figure 1C,D). RT-RPA and RT-LAMP amplicons produced using primer sets for the IBV HA gene were not detected by CRISPR-Cas12a complexed with gRNA targeting the IAV M gene (Appendix A). Likewise, RT-RPA and RT-LAMP amplicons produced using primer sets for the IAV M gene were not detected by CRISPR-Cas12a complexed with gRNA targeting the IBV HA gene (Appendix A). These data suggest that the DETECTR assay specifically detects either IAV or IBV without exhibiting cross-reactivity with the other type.

Determination of the sensitivity of influenza virus detection by the DETECTR assay. To measure the sensitivity of the assay, we lysed IAV and IBV at various PFUs per reaction with HUDSON and used the resulting lysates for RT-RPA and RT-LAMP with primer sets specific for the IAV M (Figure 3) or IBV HA gene (Figure 4). RT-RPA and RT-LAMP amplicons were incubated with CRISPR-Cas12a complexed with gRNA targeting IAV M (Figure 3) and IBV HA (Figure 4) genes and detected using an FQ-labeled reporter assay. DETECTR combined with fluorescence assays was able to detect IAV and IBV at titers as low as 1 × 10^0^ PFUs per reaction of IAV and IBV. To simulate clinical samples, IAVs and IBVs were added to human saliva, and the sensitivity of the assay with simulated samples was further determined. The DETECTR combined with fluorescence assays was able to detect IAV and IBV in human saliva at titers as low as 1 × 10^1^ PFUs (Appendix A). The sensitivity of the assay with simulated samples was reduced 10-fold.

In addition to the fluorescence assay, an instrument-free lateral flow assay was employed with DETECTR for rapid and convenient detection of influenza viruses (Figure 2 and Figure 3). RT-RPA and RT-LAMP amplicons were incubated with CRISPR-Cas12a complexed with gRNA targeting IAV M or IBV HA genes and a ssDNA-FB substrate. After incubation, the reaction mixture was applied to a paper strip and a positive result was inferred within 2 min from an increase in the T line caused by cleaved substrates (Figure 1). Data interpretation was performed at 2 min after applying samples because an increase in the T line was also detected in the control lane after 2 min, indicating the development of false-positive results with extended development (data not shown). Consistent with the fluorescence assay results, the lateral flow assay was capable of detecting as little as 1 × 10^0^ PFUs per reaction of IAV and IBV by the naked eye (Figure 3 and Figure 4). Taken together, these data indicate that DETECTR combined with fluorescence or lateral flow assays is able to detect IAV and IBV with high sensitivity and specificity.

## 3. Discussion

In this study, we developed a method (DETECTR assay) that allows specific detection of IAV or IBV at titers as low as 1 PFU per reaction within 75 to 85 min, starting from viral samples. The sensitivity of the assay was further reduced 10-fold with simulated samples, possibly due to inhibitors in human saliva. While our manuscript was in preparation, Mayuramart et al. reported that 10^3^ RNA copies of IAV or IBV per reaction could be detected using a CRISPR-Cas12a-based assay [14]. Although there were differences between viral samples employed in these two studies, the DETECTR assay developed in the current study exhibited more than a 100-fold increase in the sensitivity of detection. Future studies are warranted to verify these findings using clinical samples.

We employed RT-RPA and RT-LAMP for isothermal amplification of influenza virus RNAs and found that both methods efficiently amplified viral nucleic acids. RT-RPA and RT-LAMP primer sets designed for specific amplification of IAV M and IBV HA genes did not cross-react with the other virus type. The sensitivity of the diagnostic method was further increased though use of the DETECTR assay, and gRNAs designed for specific detection of IAV M and IBV HA genes exhibited specific detection without cross-reactivity.

For rapid, field-applicable diagnosis, a lateral flow assay was combined with the DETECTR assay. Notably, the lateral flow assay showed the same sensitivity of detection as the fluorescence assay-1 PFU per reaction. However, a drawback of the lateral flow assay was the appearance of a positive band after allowing the reaction to develop for more than 2 min. In addition, the HUDSON method for pre-treatment of viral samples has the disadvantage of requiring a thermocycler. Development of a specific, field-deployable DETECTR-based diagnostic test will require addressing the drawbacks of a false-positive band in the lateral flow assay and the requirement for a thermocycler for pre-treatment of viral samples.

## 4. Materials and Methods

Viruses. Two types of influenza viruses, A/Puerto Rico/8/1934(H1N1) and B/Brisbane/60/2008, were propagated in MDCK cells, and viral titers were determined by plaque assay as previously described [15]. Experiments with viruses were performed in a biosafety level-2 (BSL-2) laboratory. For the DETECTR assay, viral particles were lysed by HUDSON as previously described [13]. Briefly, RNase was inactivated by adding EDTA to a final concentration of 1 mM, and the reaction mixture was incubated at 95 °C for 10 min, 50 °C for 20 min and then 95 °C for 5 min using a thermocycler. To simulate clinical samples, 0.1 volume of viral particles were added to human saliva and lysed by HUDSON.

DETECTR assay. The DETECTR assay was performed using reverse transcription recombinase polymerase amplification (RT-RPA) or reverse transcription loop-mediated isothermal amplification (RT–LAMP) to amplify viral RNAs and LbCas12a (New England Biolabs, Ipswich, MA, USA) for the trans-cleavage assay. RT-RPA was performed using a TwistAmp Basic kit (TwistDx, Cambridge, UK), and RT-RPA primers were designed according to the TwistAmp Basic kit manual. Briefly, a 50 µL reaction mixture containing 1 µL sample, 0.24 µL of forward and reverse primer (10 µM each; Table 1), 29.5 µL of rehydration buffer, 11.2 µL of nuclease free water, 1 µL TOPscript Reverse Transcriptase (Enzynomics, Daejeon, Korea) and 2.5 µL of 280 mM magnesium acetate was incubated at 42 °C for 40 min. RT–LAMP was performed as suggested by New England Biolabs (www.neb.com/protocols/2014/10/09/typical-rt-lamp-protocol, accessed on 2 September 2020). RT-LAMP primers were designed using PrimerExplorer v.5 (https://primerexplorer.jp/e/, accessed on 25 August 2020) with compatible gRNAs (Table 1). LAMP primers were added to Isothermal Amplification Buffer containing 6mM MgSO_4_, 10 mM dNTP mix and WarmStart RTx Reverse Transcriptase (New England Biolabs), and the reaction mixture was incubated with input viruses at 65 °C for 30 min. RT-RPA and RT-LAMP amplicons were analyzed by electrophoresis in 2% agarose gels.

LbCas12a trans-cleavage assays were performed similarly to those previously described [11,16]. The LbCas12a-gRNA complex was created by pre-incubating 13 µL of nuclease free water, 2.5 µL of 2 µM LbCas12a, 2.5 µL of 1 µM gRNA (Table 2) and 2 µL of 10× NEBuffer 2.1 for 30 min at 37 °C. For fluorescence assays, 4 µL of RT-RPA and RT-LAMP amplicons, 20 µL of NEBuffer 2.1, 20 µL of RNA-protein complex and 4 µL of 1 µM FQ–labeled reporter (/56-FAM/TTATT/3IABkFQ/; Integrated DNA Technologies, Coralville, IA, USA) were added directly to a 96-well microplate after formation of the RNA–protein complex. The resulting reaction mixtures were incubated for 20 min at 37 °C, and fluorescence measurements were taken at the start and end of incubation (λex, 485 nm; λem, 535 nm).

For lateral flow assays, 4 µL of RT-RPA and RT-LAMP amplicons was combined with 38 µL of LbCas12a-gRNA complex, 40 µL of NEBuffer 2.1 and 2 µL of 10 µM lateral flow cleavage reporter (/56-FAM/TTATT/3Bio/; Integrated DNA Technologies). The reaction mixture was incubated for 20 min at 37 °C and applied to a lateral flow strip (Milenia HybriDetect 1, TwistDx). Results were analyzed 2 min after application.

## Figures and Tables

**Figure 1 biosensors-11-00088-f001:**
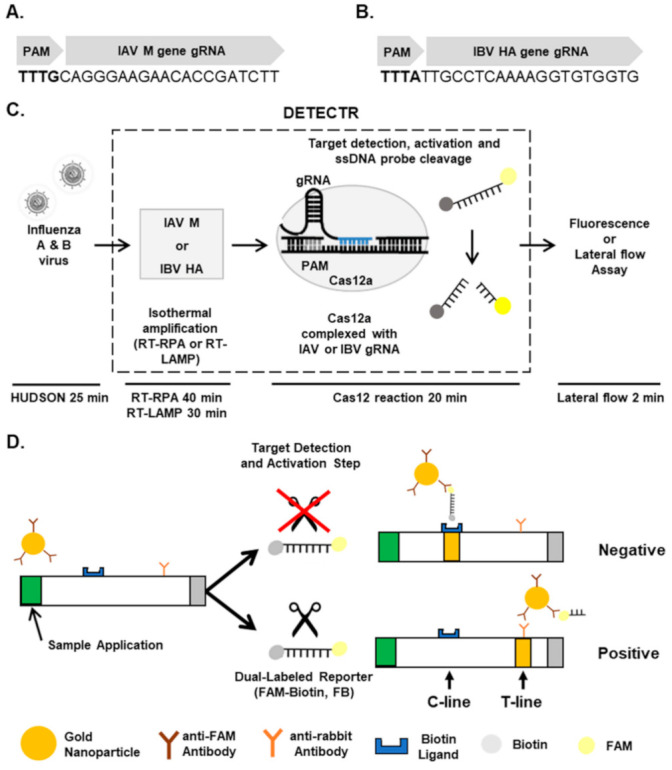
Detection of influenza virus types using CRISPR-Cas12a. (**A**,**B**) gRNAs for targeting IAV Matrix (**A**) and IBV HA (**B**) genes. (**C**) Workflow for detecting IAV and IBV by CRISPR-Cas12-based DETECTR combined with a fluorescence or lateral flow assay. (**D**) Schematic diagram showing how to distinguish between positive and negative results in a lateral flow strip. FB, FAM-Biotin; C-line, control line; T-line, test line.

**Figure 2 biosensors-11-00088-f002:**
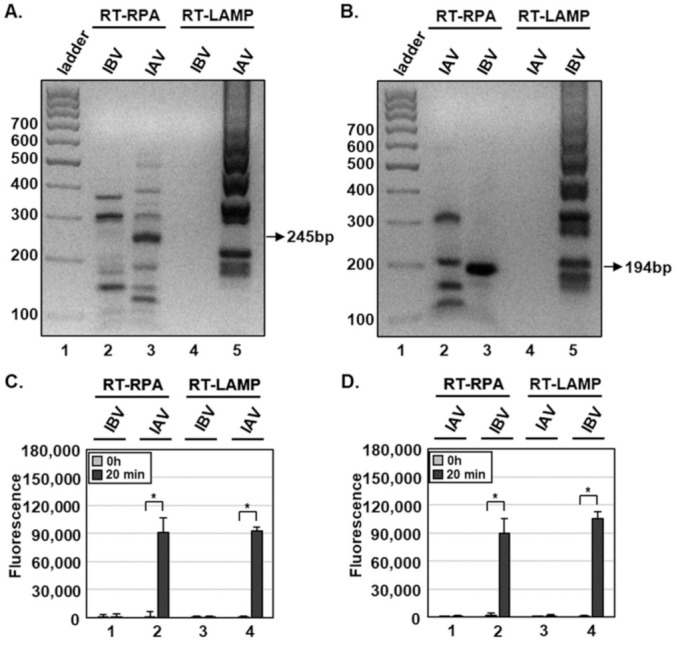
Detection of influenza viruses using DETECTR combined with fluorescence assay. (**A**,**B**) One hundred plaque forming units (PFUs) of IAV and IBV per reaction were used to amplify viral nucleic acids by either RT-RPA or RT-LAMP with primer sets specific for the IAV M (**A**) or IBV HA (**B**) gene. RT-RPA or RT-LAMP amplicons for the IAV M or IBV HA gene were visualized by gel electrophoresis. Sizes of amplicons for IAV M and IBV HA genes were approximately 245 and 194 bp, respectively. (**C**,**D**) RT-RPA and RT-LAMP amplicons were detected by DETECTR combined with fluorescence assays using gRNAs targeting the IAV M (**C**) or IBV HA (**D**) gene. Fluorescence signals of DETECTR on RT-RPA or RT-LAMP amplicons were saturated within 20 min. Values are presented as means ± s.d (error bars) (*n* = 3 replicates; * *p* < 0.05 between samples, two-sample *t*-test).

**Figure 3 biosensors-11-00088-f003:**
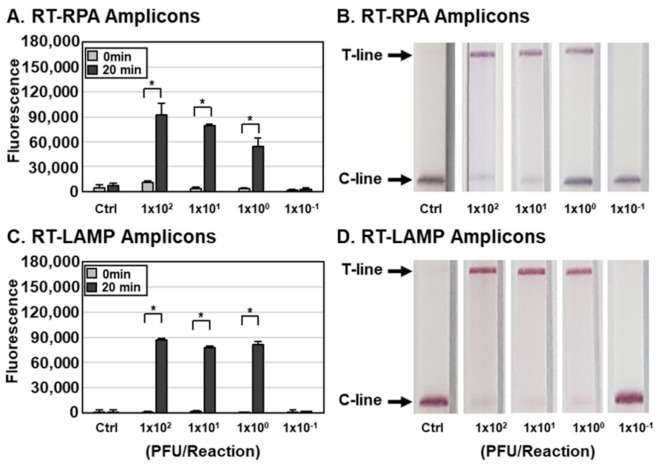
Sensitivity of the DETECTR assay for IAV. (**A**–**D**) Different concentrations of IAVs (1.0  ×  10^−1^ to 1.0  ×  10^4^ PFUs per reaction) were used to amplify viral nucleic acids using either RT-RPA (**A**,**B**) or RT-LAMP (**C**,**D**) with primer sets specific for the IAV M gene. RT-RPA and RT-LAMP amplicons were detected by DETECTR combined with fluorescence (**A**,**C**) or lateral flow (**B**,**D**) assays using gRNAs targeting the IAV M gene. The fluorescence signals of DETECTR on RT-RPA or RT-LAMP amplicons were saturated within 20 min. Lateral flow assay results were assessed 2 min after inserting the strip into the sample. Values are presented as means ± s.d (error bars) (*n* = 3 replicates; * *p* < 0.05 between samples, two-sample *t*-test). C-line, control line; T-line, test line; Ctrl, control.

**Figure 4 biosensors-11-00088-f004:**
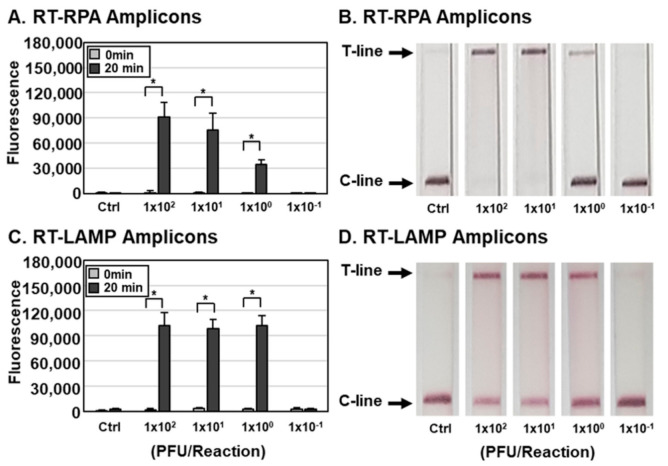
Sensitivity of the DETECTR assay for IBV. (A-D) Different concentrations of IBVs (1.0 × 10^−1^ to 1.0 × 10^4^ PFUs per reaction) were used to amplify viral nucleic acids using either RT-RPA (**A**,**B**) or RT-LAMP (**C**,**D**) with primer sets specific for the IBV HA gene. RT-RPA and RT-LAMP amplicons were detected by DETECTR combined with fluorescence (**A**,**C**) or lateral flow (**B**,**D**) assays using gRNAs targeting the IBV HA gene. The fluorescence signals of DETECTR on RT-RPA or RT-LAMP amplicon were saturated within 20 min. Lateral flow assay results were assessed 2 min after inserting the strip into the sample. Values are presented as means ± s.d (error bars) (*n* = 3 replicates; * *p* < 0.05 between samples, two-sample *t*-test). C-line, control line; T-line, test line; Ctrl, control.

**Table 1 biosensors-11-00088-t001:** Primers for RPA and LAMP.

Application	Virus	Gene	Primer	Sequence
RPA	IAV	M	IAV-M-F	AAGATGAGTCTTCTAACCGAGGTCGAAACG
			IAV-M-R	TGGACAAAGCGTCTACGCTGCAGTCCTCGC
	IBV	HA	IBV-HA-F	GTTGATTACATGGTGCAAAAACCTGGGAAAA
			IBV-HA-R	CCTGTGTAGTAAGGCTTGCTTTTGTTTAATCCACC
LAMP	IAV	M	IAV-M-F3	TCTTCTAACCGAGGTCGAAAC
			IAV-M-B3	GTCTACGCTGCAGTCCTC
			IAV-M-FIP	AGACATCTTCAAGTCTCTGTGCGAGTACGTTCTCTCTATCATCCCGT
			IAV-M-BIP	AGACAAGACCAATCCTGTCACCTCGGTGAGCGTGAACACAA
			IAV-M-LF	TCTCGGCTTTGAGGGGG
			IAV-M-LB	CTGACTAAGGGGATTTTAGGAT
	IBV	HA	IBV-HA-F3	CAAATCAAACAGAAGACGGA
			IBV-HA-B3	GCTTTTGTTTAATCCACCGT
			IBV-HA-FIP	ACCCCTTTGATAGGTAATTGTTCCAAAGTGGTAGAATTGTTGTTGA
			IBV-HA-BIP	CAAGTGGCAGGAGCAAGGTAATTTTTCGTGGAGGCAATC
			IBV-HA-LF	GTTTTCCCAGATTTTTGCACCATG
			IBV-HA-LB	GGATCCTTGCCTTTAATTGGAGAAG

**Table 2 biosensors-11-00088-t002:** gRNA sequences.

Virus	Gene	gRNA	Sequence	PAM
IAV	M	gRNA-IAV-M	CAGGGAAGAACACCGATCTT	TTTG
IBV	HA	gRNA-IBV-HA	TTGCCTCAAAAGGTGTGGTG	TTTA

## Data Availability

All data and materials supporting the conclusions are described and included in this manuscript.

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
