# Peer review of "Specific Detection of Influenza A and B Viruses by CRISPR-Cas12a-Based Assay"

_biosensors, 2021, doi:10.3390/bios11030088_

Round 1
Reviewer 1 Report
The authors developed a CRISPR-Cas12a-based assay for the detection and differentiation of influenza A and B viruses which allows the specific detection of these viruses within a short period of time displaying a high sensitivity. Although there are similar assays which have been reported the work is of interest.
Minor comments.
Title
please include A and B because you do not investigate other types
Introduction
lines 42 and 43: please focus more on a quick application of this assay for diagnostics rather than for monitoring influenza epidemics and pandemics because the latter needs much more (differentiation of subtypes and lineages as well as genetic, antigenetic and resistance characterization); because A and B viruses can circulate in parallel the application of such an assay could be especially important in hospitals to allow decisions for separation of infected patients
Results
lines 73-74 second sentence under 2. Results: "unextracted diagnostic samples" is twice in the sentence
Author Response
Dear Ms. Wang,
Please find below our response to the reviewers for the manuscript #Biosensors-1127009 (Title: Specific detection of influenza virus types by CRISPR-Cas12a-based assay) item by item. We highly appreciate the reviewers’ comments and your assistance.
Best regards,
Yoon-Jae Song, Ph.D.
Reviewer #1:
The authors developed a CRISPR-Cas12a-based assay for the detection and differentiation of influenza A and B viruses which allows the specific detection of these viruses within a short period of time displaying a high sensitivity. Although there are similar assays which have been reported the work is of interest.
Point 1: Please include A and B because you do not investigate other types.
Response 1: As suggested by the reviewer, the title has been changed to “Specific detection of influenza A and B viruses by CRISPR-Cas12a-based assay”.
Point 2: lines 42 and 43: please focus more on a quick application of this assay for diagnostics rather than for monitoring influenza epidemics and pandemics because the latter needs much more (differentiation of subtypes and lineages as well as genetic, antigenetic and resistance characterization); because A and B viruses can circulate in parallel the application of such an assay could be especially important in hospitals to allow decisions for separation of infected patients
Response 2: As suggested by the reviewer, a sentence in lines 47-49 has been revised as follows: “Therefore, a rapid diagnostic test to distinguish between IAV and IBV infections is necessary for preventing the spread of influenza viruses.”
Point 3: lines 73-74 second sentence under 2. Results: "unextracted diagnostic samples" is twice in the sentence
Response 3: HUDSON is an abbreviation for heating unextracted diagnostic samples to obliterate nucleases. The sentence in lines 78-79 was revised as follows: “Briefly, unextracted diagnostic samples of IAVs and IBVs were lysed by the method known as heating unextracted diagnostic samples to obliterate nucleases (HUDSON).”
Reviewer 2 Report
Park et al. apply CRISPR-associated (Cas)-based assay for IAV and IBV diagnostics. They report a resolution of detection at 1 x 100 plaque forming units (PFUs) per reaction and the assay is proposed to be 100 fold more sensitive than an existing CRISPR Cas-based assay for IAV/IBV detection. The assay was also adapted to a lateral flow stripassay that does not require additional analytic equipment. The method is presented as a diagnostic test to rapidly distinguish IAV and IBV infections.
General
The work is well executed. Rapid more sensitive diagnosis for IAV and IBV may help meet and urgent need and are thus of important interest.
Specific
Authors suggest that their DETECTR assay is 100 fold more sensitive to a Cas12a-based IBV and IAV detection assay published last year, but note that there were differences between viral samples of these two studies. Sample preparation (eg. HUDSON etc) can have an important impact on detection. Authors should comment here.
Other:
In abstract “CIRSPR” should be “CRISPR
Author Response
Dear Ms. Wang,
Please find below our response to the reviewers for the manuscript #Biosensors-1127009 (Title: Specific detection of influenza virus types by CRISPR-Cas12a-based assay) item by item. We highly appreciate the reviewers’ comments and your assistance.
Best regards,
Yoon-Jae Song, Ph.D.
Reviewer #2:
Park et al. apply CRISPR-associated (Cas)-based assay for IAV and IBV diagnostics. They report a resolution of detection at 1 x 100 plaque forming units (PFUs) per reaction and the assay is proposed to be 100 fold more sensitive than an existing CRISPR Cas-based assay for IAV/IBV detection. The assay was also adapted to a lateral flow stripassay that does not require additional analytic equipment. The method is presented as a diagnostic test to rapidly distinguish IAV and IBV infections.
The work is well executed. Rapid more sensitive diagnosis for IAV and IBV may help meet and urgent need and are thus of important interest.
Point 1: Authors suggest that their DETECTR assay is 100 fold more sensitive to a Cas12a-based IBV and IAV detection assay published last year, but note that there were differences between viral samples of these two studies. Sample preparation (eg. HUDSON etc) can have an important impact on detection. Authors should comment here.
Response 1: To address the issue, we simulated clinical samples by adding IAVs and IBVs to human saliva and performed the assay with simulated samples. We found that the assay was able to detect IAV and IBV in human saliva as low as 1X101 PFUs (The sensitivity of the assay with simulated samples was reduced 10-fold). These data have been included in the manuscript as supplementary figure S2. Lines 112-116, 136-138 and 166-167 have been revised to describe and discuss these data.
Point 2: In abstract “CIRSPR” should be “CRISPR
Response 2: A typo has been corrected.
Reviewer 3 Report
Manuscript by Bum Ju Park et al. concerns developing of diagnostic test for detection of influenza virus A and influenza virus B, which combine RT-RPA and RT-LAMP and is based on CRISPR-Cas12a DNA endonuclease-targeted CRISPR trans-reporter (DETECTR). Manuscript could be interesting for potential readers when the method will be more improved. I am recommending rejection of the manuscript and consider for publication after manuscript will be significantly improved.
My details remarks are as follow:
1/ in the manuscript is lack of experimental details. It is impossible to repeat experiments based on present form of manuscript.
2/ there is lack of evaluation of described method on real patient samples or on model samples that contain mixture of other RNA (total human RNA). This test is highly important to show specificity of detection where percentage of viral RNA in sample is very low.
Author Response
Dear Ms. Wang,
Please find below our response to the reviewers for the manuscript #Biosensors-1127009 (Title: Specific detection of influenza virus types by CRISPR-Cas12a-based assay) item by item. We highly appreciate the reviewers’ comments and your assistance.
Best regards,
Yoon-Jae Song, Ph.D.
Reviewer #3:
Manuscript by Bum Ju Park et al. concerns developing of diagnostic test for detection of influenza virus A and influenza virus B, which combine RT-RPA and RT-LAMP and is based on CRISPR-Cas12a DNA endonuclease-targeted CRISPR trans-reporter (DETECTR). Manuscript could be interesting for potential readers when the method will be more improved. I am recommending rejection of the manuscript and consider for publication after manuscript will be significantly improved.
Point 1: in the manuscript is lack of experimental details. It is impossible to repeat experiments based on present form of manuscript.
Response 1: Somehow, tables for primers and qRNAs were not included in the manuscript. The manuscript has been revised to include tables.
Point 2: there is lack of evaluation of described method on real patient samples or on model samples that contain mixture of other RNA (total human RNA). This test is highly important to show specificity of detection where percentage of viral RNA in sample is very low.
Response 2: As suggested by the reviewer, we simulated clinical samples by adding IAVs and IBVs to human saliva and performed the assay with simulated samples. We found that the assay was able to detect IAV and IBV in human saliva as low as 1X101 PFUs (The sensitivity of the assay with simulated samples was reduced 10-fold). As anticipated, the assay was hindered by inhibitors present in human saliva. These data have been included in the manuscript as supplementary figure S2. Lines 112-116, 136-138 and 166-167 have been revised to describe and discuss these data.
Round 2
Reviewer 3 Report
Manuscript by Bum Ji Park et al. was greatly improved according to reviewer comments. Additional data and discussion makes the manuscript clear, interesting and valuable for potential readers. After these significant changes I recommend to accept manuscript.